# Research on the Effects of Soundscapes on Human Psychological Health in an Old Community of a Cold Region

**DOI:** 10.3390/ijerph19127212

**Published:** 2022-06-12

**Authors:** Peng Cui, Tingting Li, Zhengwei Xia, Chunyu Dai

**Affiliations:** 1School of Landscape Architecture, Northeast Forestry University, Harbin 150040, China; cuipeng@nefu.edu.cn (P.C.); may_lting@nefu.edu.cn (T.L.); daichunyu0629@nefu.edu.cn (C.D.); 2School of Architecture, Soochow University, Suzhou 215123, China

**Keywords:** soundscape, physical health, acoustic comfort, sound preference, old community

## Abstract

The acoustic environment of residential areas is critical to the health of the residents. To reveal the impact of the acoustic environment on people’s mental health and create a satisfactory acoustic setting, this study took a typical old residential area in Harbin as an example, conducted a field measurement and questionnaire survey on it, and took typical acoustic sources as the research object for human body index measurement. The relationship between heart rate (HR), skin conductivity level (SCL), physiological indicators, semantic differences (SD), and psychological indicators was studied. The sound distribution in the old community was obtained, determining that gender, age, and education level are significant factors producing different sound source evaluations. Music can alleviate residents’ psychological depression, while traffic sounds and residents’ psychological state can affect the satisfaction evaluation of the sound environment. There is a significant correlation between the physiological and psychological changes produced by different sounds. Pleasant sounds increase a person’s HR and decrease skin conductivity. The subjects’ HR increased 3.24 times per minute on average, and SCL decreased 1.65 times per minute on average in relation to hearing various sound sources. The SD evaluation showed that lively, pleasant, and attractive birdsongs and music produced the greatest HR and SCL changes, and that the sound barrier works best when placed 8 m and 18 m from the road.

## 1. Introduction

A “healthy building” is a new requirement in modern society to ensure functionality and the quality of architecture. For this, concept and practice must break through the shackles of traditional disciplines to develop towards achieving multi-discipline and cross-field integration. The acoustic environment of buildings and cities is an essential part of the quality of healthy buildings. As such, the impact of the acoustic environment on human health and the corresponding soundscape design has been the focus of research in recent years [1]. The acoustic environment directly impacts the health of residents, especially in old communities, making it an important part of the urban renewal policy of China.

The old communities are always located in the city center with spatial openness, and they are accompanied by diverse sound sources, such as road traffic noises and the cries of children. Due to the lack of focus on effective community planning, cars are parked in a disorderly fashion, and illegal buildings are constructed, making it hard to dissipate the noise. Moreover, the greening rate of old communities is relatively low. It is difficult to create natural sounds, such as wind blowing leaves and animal sounds, such as birds singing and cicadas chirping. With the development of the current society, people are unsatisfied with outdoor acoustic comfort.

The noise of the living environment may lead to cardiovascular and cerebrovascular diseases, sleep problems, irritability, and cognitive disorders in children [2,3,4]. Exposure to traffic sounds has been associated with an increased risk of adverse health outcomes, both physiologically and psychologically [5]. Bergomi et al. [6] conducted physiological tests on students, showing that high decibel noise could affect the neuroendocrine system of the human body and damage sensory function. However, most of the earlier studies only focus on the negative factors of sound instead of considering the physiological effects of the positive impact of sound. In recent years, soundscape research has gradually become independent of noise research. The focus of research has shifted from the adverse effects of environmental sounds to their overall impact, taking human perception into account and treating sound as a possible resource for promoting health and supporting the environment [7]. The ISO 12913-1 standard defines “soundscape” as “the acoustic environment perceived, experienced, and/or understood by an individual or group of people in a particular setting” [8].

The association between positive soundscape perception (e.g., happiness, calmness, etc.) and positive health effects (e.g., increased recovery rates, reduced stress-induced mechanisms, etc.) is one of the key questions in soundscape research. A two-dimensional emotional assessment was carried out for 60 groups of sounds and their corresponding pictures, and the changes in ECG, EEG, and skin resistance induced by these sounds were recorded. It was found that the physiological changes were highly correlated with the results of self-emotional evaluation [9]. Chuen et al. [10] studied the influence of changes in single parameters of sound on physiological indicators. They analyzed the effect of sound on heart rate (HR), skin resistance, respiratory rate, and facial muscles. The results show that changes in all sound parameters lead to an increase in HR. The response of the skin’s electrical signals is affected by changes in timbre, intensity, and rhythm, among which respiratory rate is susceptible to changes in rhythm. Blood [11], Schmidt [12], and Sammler [13] find that happy and exciting music clips were associated with electrical activity in the left frontal lobe. In contrast, fearful and sad music increased electrical activity in the right frontal lobe. Meng et al. [14] revealed the relationship between low-frequency noises and fatigue in the working environment. Kang [15] found that the physiological response of human beings was strongly affected by the measurement time and soundscape type and the relationship between the physiological signal and the subjective restorative scale.

The attention restoration theory suggests that nature (such as exposure to natural sounds, like the sound of waterfalls) improves cognition and helps with recovering from stress [16,17,18,19]. By reviewing literature with the keywords soundscape, health, and quality of life, 130 research papers were retrieved which supported that positive soundscapes were significantly associated with self-reported good health. The primary method of soundscape physiology research is an experimental design based on passive listening and event-related or stimulus locking. The experimental subjects are mainly homogenous. The physiology and neurophysiology of the soundscape are still in their infancy, and there are many aspects worthy of further study.

Old communities are a part of urban modernization, with Chinese characteristics. Old communities carry generational information, such as unique perspectives and traditions, though they face many health challenges brought about by rapidly changing urban environments and lifestyles. According to environmental recovery theory [20], an appropriate setting can promote the recovery of individuals within it from a consumption state; otherwise, consumption will increase. Therefore, this paper studies the correlation between the soundscape of old communities and human physiological health. Then, according to the analysis, the study obtains sound sources with positive effects on human health to provide a reference point for improving the sound environment in reconstructing old communities.

## 2. Methodology

### 2.1. Study Area and Measurement of dB(A)

Harbin (125°420–130°440 E longitude, 10°040–46°400 N latitude) is the capital of Heilongjiang Province, China. The city is characterized by long winters, short and dry summers, and relatively short spring and autumn seasons. The present study selects the typical representative old residential areas of the Songshan community (Site A) and the Liaohe community (Site B) in Harbin as the research fields. The layout of Site A possesses a peripheral and determinant architecture, and the layout of Site B is an enclosed mode. As Figure 1 shows, the measuring points were arranged in a network format. P1, P2, P11, and P12 were located in the exterior walkway to test the sound pressure level (SPL) of the arterial street. P7, P9, and P17 were located inside the community to focus on the impact of noise on multi-story buildings. P8 and P18 were set in the square to focus on the effects of square dancing. Another test point was established to study the influence of different architectural forms on the acoustic environment.

An AWA5680 sound level meter with a measurement range of 30–130 dB(A) was selected for this study [21]. According to the Acoustic Environmental Noise Measurement Method (GB/T 3222-94), the three daytime periods of 8:00–10:00, 14:00–16:00, and 19:00–21:00 were selected for measurement. In this test, the sound level meter was set to count every 10 s. Each point was continuously tested for 10 min, and each measuring point was tested for a total of 12 times. The distances between the measuring point, the reflector, and the outer wall were greater than 3.5 m and 1 m, respectively, and the probe height was set at 1.5 m (Figure 2a).

### 2.2. Questionnaire Design

To explore the acoustic environment characteristics of the old residential area and residents’ preference for each acoustic source, a questionnaire was used to obtain the psychological data of the subjects with the following questions: the sound source, degree of preference for the sound source, the social background of the subjects, and psychological perceptions. The PHQ-9 depression scale was used to obtain psychological perceptions.

In the questionnaire, the attitudes of the subjects were measured using a five-point Likert-type scale (Table 1), which has been widely used in survey research on the environmental effects of subjective comfort [22]. A total of 300 residents were surveyed from September 2021 to March 2022. The reliability coefficient of the questionnaire was estimated at 0.87 (Cronbach’s alpha). The KMO coefficient was 0.861, and Bartlett’s spherical test results were significant (*p* < 0.001).

### 2.3. Subjective Evaluation of Sound Perception

Electrodermal activity (EDA) and HR are widely used to assess the physiological response to sound stimuli and are suggested as sensitive indicators for evaluating the impact of sounds [23,24,25]. The EDA was measured using two electrodes (HKR-11, range: 100 to 2500 kΩ; accuracy: 2.5 kΩ; sample frequency: 50 Hz) attached to the subject’s index and middle fingers of the non-dominant hand. The HR was measured by a photoplethysmography (PPG) sensor (HKG-07, range: 30 to 250 bpm; accuracy: 1 bpm; sample frequency: 16 Hz) attached to the ring finger [26].

According to the feasibility of the experiment, the requirements of sample size for data analysis, and to ensure that the subjects had good hearing conditions, the subjects of this study were determined to be young people (average age: 30) with high hearing sensitivity [27]. Thirty residents were selected to voluntarily join the test in NEFU’s building physics laboratory. They were then informed of the purpose and process of the experiment. All participants had normal hearing, were healthy, and did not take prescription drugs. The physiological environment was in a stable state (temperature = 21–23 °C; background SPL < 25 dBA).

Participants were asked to wear blindfolds during the experiment and listen intently using headphones (Figure 2b). White noise was played first, and other audio was played randomly. Each audio lasted two minutes, and the interval between each sound was 60 s [9,28]. The subjects were required to fill in the simple mood state scale to evaluate their mental state, and the degree of pleasure and excitement were used as evaluation items during the interval time.

The semantic difference method was used to obtain residents’ evaluation of different sound sources [29]. Participants’ psychological ratings were obtained by asking them to choose their feelings on a verbal scale. The SD method selects multiple pairs of adjectives to represent the psychological feelings of the subjects towards the evaluated objects, which is beneficial for the quantitative evaluation. This study refers to the adjective pairs commonly used in the existing research and summarizes the evaluation items into 11 items. Participants were asked to rate the sounds they heard in the audio using the five-point Likert scale (Table 2).

## 3. Results

### 3.1. SPL Distribution of the Study Area

Figure 3 shows the SPL distribution of the study area. It can be seen from Figure 3 that the old multi-story residential area is seriously disturbed by noise far beyond China’s acoustic environmental quality standards (55 dB). Table 3 shows the maximum value of SPL during the different periods in the study area. As the table shows, during 8:00–10:00, the points along the street (P1, P2, P3, P4, P12, P13) appear to have a higher SPL, and the maximum interpolation reached 8.3 dB(A) among the points, which is similar during 19:00–21:00. This is because traffic noises dominate the acoustic environment during these periods. It is worth noting that the points on the square (P8, P9) appear to have a higher SPL during 19:00–21:00 because square dancing affects the acoustic environment during this time. The SPL distribution inside the community during 14:00–16:00 differs slightly, but the maximum interpolation reaches 24.1 dB(A) between the point inside the community and outside the road. The average SPL in Site A is lower than in Site B, concluding that the enclosed layout is superior to the determinant layout in noise reduction. It is worth noting that there are two 1-story illegal buildings around the measuring point 13–16. In the whole day SPL distribution, these buildings played a certain role in hindering the propagation of external traffic noise. The distribution of SPL values is 1.8 dB lower than that of the surrounding 8-story building area. Therefore, it can be seen that low-rise small buildings have a certain impact on the SPL distribution of the residential area.

### 3.2. Evaluation of the Sound Environment Based on the Sound Types and Sources

The proportion of women in the questionnaire was 58.1%. Residents aged 31–64 accounted for 52.9%, and those over 65 accounted for 11.0%. Those with at least a middle school education level accounted for 65.1% of the subjects, and uneducated people accounted for 4.1%. In terms of working conditions, the highest proportion of residents with other occupations was 48.8%, and the unemployed accounted for 12.3% of the subjects. In terms of income, people with an income of CNY 1500–3000 accounted for 43.6%, and those with an income greater than CNY 5000 accounted for 12.8%. The monthly income of residents in the sample community is relatively low.

Figure 4 shows the frequency distribution of community residents hearing different sound sources in different seasons. People are sensitive to the sounds of traffic, animals, and music. In addition, in evaluating the outdoor acoustic environment, the proportion of basically satisfied and very dissatisfied was 67.9% and 5.8%, respectively. Most of the subjects could adapt to the existing acoustic environment.

Table 4 shows the types of sound sources heard by the subjects in the questionnaire and the statistical analysis of their subjective evaluation. As the table shows, the most popular sounds were birdsongs (4.10) and the sound of rustling leaves (3.94), followed by musical instruments playing (3.58) and music (3.55), then the square dancing (3.23), indicating that residents preferred music and natural sounds. The most disliked sounds were factory machinery (1.61) and construction sounds (1.69), followed by the sound of a device running (1.82) and thunder (2.15), followed by tires/road traffic (2.05) and car horns (2.14). It is worth noting that the evaluation value of the sound of firecrackers is 2.81, but only 53% of the subjects like it because this sound only appears during the Spring Festival. Additionally, the evaluation value of the sound of children playing is 3.15, contrary to normal expectations. By analyzing the questionnaire, it is found that the residents with children gave a high evaluation, but most of the residents without children in the family cannot tolerate it. For the traffic noise inside the residential area, the subjects indicated that it was within the acceptable range. This was because the sound of internal driving vehicles was only an occasional sound source. At the same time, the internal roads in the old community were narrow, and the driving speed of vehicles was slow. The generated sound had no significant impact on the residents. However, the traffic noise of the external lane is unacceptable to the residents (which result is the same as that in previous studies), because there is a significant sound sequence phenomenon in the external lane; also, the vehicle speed is fast and the SPL is high, [30].

### 3.3. Correlation between Acoustic Evaluation and Sound Sources

SPSS was used to process the questionnaire data. It was found that the age, educational background, occupation, and other social characteristics of interviewees had a particular impact on the evaluation of sound preference, which was similar to previous research results [31]. Figure 5 shows the correlation analysis results of subjects’ social characteristics, satisfaction with the acoustic environment, psychological state, and sound sources.

Regarding individual characteristics, the correlation between gender and the sound of firecrackers and dogs barking was 0.307 * and 0.332 *, respectively (*p* < 0.05), where men prefer these two sounds over women. The correlation between age and the sounds of footsteps, rain, and dogs barking was −0.369 *, −0.347 *, and −0.369 *, respectively (*p* < 0.05), where these sounds were less likely to be enjoyed by older residents. The correlation between education level and the sound of chatting, rain, thunder, and cicadas was 0.475 ** (*p* < 0.01), 0.355 *, 0.361 *, −0.380 *, respectively (*p* < 0.05), signifying that with the improvement of education level, the residents’ tolerance to the sounds of voices chatting and thunder is higher. In comparison, the tolerance to the sound of cicadas is lower. The preference for the sound of rain is higher. The correlation between the residential floor and the sound of construction is 0.328 * (*p* < 0.05), and the correlation between the location of the residence and the sound of road cleaning is −0.475 ** (*p* < 0.01), signifying that the higher the residential floor is, the higher the degree of tolerance to the sound of construction. Furthermore, residents directly facing the street have a higher degree of tolerance to the sound of roads being cleaned.

In terms of acoustic environment satisfaction, the correlation between outdoor acoustic environment satisfaction and traffic sounds and residents’ psychological depression was −0.379 * (*p* < 0.05) and −0.530 **, respectively (*p* < 0.01). This shows that residents exposed to increased traffic noises or suffering from psychological depression have lower satisfaction with the outdoor acoustic environment. The correlation coefficient between residents’ psychological depression and music was 0.489 ** (*p* < 0.01), indicating that residents with high psychological depression liked music more.

### 3.4. Correlation Analysis between Sound Perception and Psychology

According to the questionnaire results, it is found that the sound of traffic has the most significant influence on residents, and residents generally like the sound of birds, nature sounds, and music. Therefore, the sound sources of the study are set as the sound of traffic, the sound of birds, the sound of wind blowing on leaves, and the sound of music. A white noise was established as the control group. Except for the music sound, other sound sources were extracted from the on-site recordings collected in the test area by a voice recorder (SONY PCM-M10) and calibrated according to ISO 1996-2:2017. To avoid the interference of different noises, the sound sources were collected at relatively quiet times throughout the day (Recording time: 8:00–10:00, 14:00–16:00, 19:00–21:00) [32,33]. The sampling audio format was set to WAV, dual-channel 16 Bit, and the sampling frequency was 44.1 kHz [34]. Adobe Design CS6 software (Adobe Co., New York, NY, USA) extracted 2-min snippets from the field recordings as the experimental sound source. Based on field measurement, the average SPL of the test site was 59 dB(A), so the A-weighted equivalent sound level was set to 60 dB(A).

Figure 6 shows the HR changes after hearing different types of sound stimulation. As the figure shows, the changes in HR after hearing birdsongs, rustling leaves, music sounds, and traffic sounds were 5.41%, 3.90%, 4.18%, and 2.18%, respectively. Compared with the control group, the increase in HR indicates increased sympathetic nerve activity, to some extent. The greater the increase in HR, the more the residents tended to be excited or happy. According to the HR changes for different types of sounds, birdsongs, rustling leaves, and music have positive effects. Here, birdsongs showed the strongest effect, whereas the sound of wind blowing on leaves had the weakest effect.

The statistical analysis of the changes in SCL for various sound types found that after listening to the control group and then listening to other sound sources, the SPL appeared to have a certain decreasing trend, except for traffic sounds. The SCL changes after hearing birdsongs, music sounds, rustling leaves, and traffic sounds are −63.30%, −51.55%, −28.66%, and 7.84%, respectively (Figure 7). It was found that birdsongs strongly influenced the human body’s mood of pleasure or relaxation, followed by music, and finally, by the sound of rustling leaves. The sound of traffic did not produce pleasure or relaxation.

Figure 8 shows the evaluation scores of pleasure and excitement changes under different sound source backgrounds. As Figure 8 shows, the evaluation of the pleasure and excitement degrees for traffic sounds is negative, indicating that people reject the urban noise environment. The remaining recordings were rated positively, suggesting that such sounds brought people a distinct sense of pleasure and excitement. There were differences in evaluating the degrees of pleasure and excitement among different sounds. The scores of the degrees of pleasure and excitement for birdsongs (1.64 and 1.52) and music (0.79 and 0.82) were higher. The scores for wind blowing on leaves were the lowest, which were 0.73 and 0.34, respectively, indicating that all three kinds of sound sources positively affected human psychology. The effects of birdsongs and music are more pronounced. As Table 5 shows, the results show that HR and SCL were correlated with pleasure and excitement at *p* < 0.01, indicating that physiological indexes could comprehensively reflect the psychological changes of the human body, to a certain extent. There was a significant positive correlation between HR and the degrees of pleasure and excitement, and a significant negative correlation between SCL and the degrees of excitement and pleasure. In summary, when the human body is in a state of pleasure and excitement, the HR will show an increasing trend, while the SCL will show a trend of decline.

Table 6 shows the SD scores for different types of sounds. The table shows that the prominent evaluation item of the birdsong type is clear and lively, making people feel peaceful and happy. From the evaluation results, people’s evaluation of birdsongs is entirely positive, which is related to the special vocalization mode of birds, which meets people’s expectations of natural sounds. The evaluation item of the music sound type is clear and lively, attractive, and pleasant. People’s general feeling for music is that the sound environment is complex and smooth. This kind of psychological sense is related to the type of music, which can relieve people’s moods. The evaluation item of the sound type of rustling leaves is quiet and natural, which can provide a good sound environment and make people feel peaceful. According to the evaluation results, the three kinds of acoustic sources evaluated are all positive, providing a comfortable acoustic environment for residents.

### 3.5. Influence of Sound Barrier on SPL Distribution in an Old Community

For vertical SPL distribution, Cadna/A simulation software (DataKustik Co., Gilching, German) was used to develop an optimization strategy. The simulation parameter was set according to ISO9613. The simplified model was set to a 6-floor building (length: 40 m, width: 12 m) that was 8 m from a four-lane road, characterized by a road width of 12 m. The traffic situation was set using real-life conditions (speed: 40 km/h, number of cars: 1665/h, number of trucks: 85/h). The noise barrier was 3 m high and the same length as the road. The simulation result is shown in Figure 9.

As can be seen from Figure 9, the sound barrier effect is evident on the first floor of the building, but the overall noise reduction effect decreases with the increase in the floor height. By comparing the addition of a sound barrier at the side of the road and the addition of a sound barrier at 2 m on the side of the road, it can be seen that the location of the sound barrier has no influence on the 1st floor, but has a great impact on the 2nd floor. The sound barrier is placed closer to the road for a better sound insulation effect, and its range of influence is expanded from 2 floors to 5 floors. From the perspective of sound insulation, it is more advantageous to place the sound insulation screen on the side of the road (Table 7).

As can be seen from Table 8, with the increase in road distance, the transverse comparison SPL of each floor gradually decreases. It is found that when the length increases from 8 m to 18 m, the SPL of the 1st floor decreases by 4 dB, and that of the top floor decreases by 2 dB. When the distance drops to 58 m, the SPL of the 1st floor decreases by 11 dB, and that of the top floor only reduces by 6 dB. It can be seen that the 1st floor is most sensitive to distance. When the distance is 8 m from the road, the SPL weakens with the increase in floor level, whereas the SPL is slightly weakened on the 5th floor. When it is 18 m from the road, the SPL decreases with the increase in the number of floors. When it is 28 m from the road, the SPL increases with the increase in the number of floors because the traffic noise influences the building distance to the road, and it is related to the length of the road. By comparing the difference in the total SPL of each setback distance, it is found that the road setback distances of 18 m and 28 m are the “overall optimal,” which are reduced by 13 dB(A) compared to the previous standard.

As shown in Table 9, with the increase in the setback distance, the effect of the sound barrier also increases on the whole. However, from the perspective of the difference value, the impact of the sound barrier is 14 dB when the road distance is 8 m and 7 dB when the road distance is 18 m. Then, the effect weakens with the increase in distance and does not rebound until 48 m. Therefore, 8 m and 18 m are ideal distances for the sound barriers.

## 4. Discussion

Based on the findings of the above study, the acoustic environment of old communities seriously affects the psychological health of residents. This paper obtains the subjects’ perceptions of the existing acoustic environment, and puts forward the corresponding improvement strategies based on these perceptions. However, this paper takes the old community as the carrier, which is different from other research carriers; due to the impact of the built-up environment, it can only regulate the background sound and noise. Through the construction of a sound barrier made of trees, the ecological environment can be improved, increasing the residents’ exposure to birdsong. Factors affecting sound comfort, such as firecrackers and children’s noises, can be improved through property management and control.

A large number of scholars have done relevant research on the impact of traffic noise on residential areas. However, this study found that traffic noise is not the most intolerable sound source for residents in old communities, and mechanical noise has a greater impact on residents’ satisfaction. At the same time, the influence of traffic sounds on heart rate is not significant, and the influence on skin conductivity is the most significant. In terms of acoustic environment satisfaction and acoustic preference, Kang J et al. [35] conducted similar research on high-rise residential areas in the same region. Through comparison, it is found that the maximum SPL difference in the old community (24.1 dB) is significantly higher than that in the high-rise residential area (21 dB). At the same time, the residents’ perception of some sound sources is different from that in the modern high-rise community. For example, the satisfaction value for traffic noise in the high-rise residential area (1.8) is lower than that in the old community (2.1), indicating that the residents in the old community have adapted to the impact of traffic noise, to a certain extent. As for the value of satisfaction regarding decoration noise, the residents in old communities report lower values (1.6) than those in high-rise modern residential areas (2.2). There are few decoration sound sources in old communities, so residents are more sensitive to such sound sources, while decoration behavior in modern residential areas is more common, and residents adapt to such sound sources to a certain extent. It can be seen from the above that physiological adaptability is an important factor in the study of acoustic environment. The overall satisfaction of residents in modern residential areas is higher than that in old communities. By evaluating the comparative study item by item, it is found that the visual factor of greening is also an important factor affecting sound perception.

## 5. Conclusions

In this study, it was found that the sound environment impacts the human body index and the psychological state. The physiological response of humans was strongly affected by the measurement time, soundscape type, and the relationship between the physiological signal and subjective restorative scale. The differences between the physiological indicators and the correlation between the physiological indicators and subjective evaluation factors were determined by performing variance and canonical correlation analyses. From the obtained results, the following conclusions can be drawn:(1)The internal noise distribution in the old community has obvious time distribution characteristics. During rush hour, the maximum SPL is 91.5 dB(A) every day, which is mainly affected by external traffic noise. The maximum SPL is 84.1 dB(A) throughout the remainder of the day, which is affected by the internal living noise. These values exceed the healthy SPL range of 45–55 dB.(2)Residents are sensitive to the sound of traffic, animals, and music and have an apparent preference for all kinds of sound sources in the area (nature sounds > animal sounds > man-made sounds > commercial sounds > traffic sounds). Gender, age, and education level are significant factors that produce a different sound source evaluation. Music can alleviate residents’ psychological depression, while traffic sounds and residents’ psychological state can affect the satisfaction evaluation of the sound environment.(3)There is a significant correlation between the physiological and psychological changes for the production of different sounds. Pleasant sounds increase a person’s HR and decrease SCL. Between hearing various sound sources and silence, the subjects’ HR increased 3.24 times per minute on average, and SCL decreased 1.65 times per minute, on average. When hearing birdsongs, the most apparent change was that HR increased 4.47 times per minute, while SCL decreased 3.07 times per minute. The SD evaluation showed that lively, pleasant, and attractive birdsongs and music produced the greatest HR and skin conductivity changes.

The sound environment of the old, multi-story residential area cannot meet the living needs of residents. The acoustic simulation analysis shows that the closer the sound barrier is to the road, the better the sound insulation effect. Sound insulation is best when the barrier is located between 8 m and 18 m from the road, but the noise reduction effect is not ideal when the barrier borders green belts. The sound environment can be improved with sound barrier design and by playing background music or simulated natural sounds.

## Figures and Tables

**Figure 1 ijerph-19-07212-f001:**
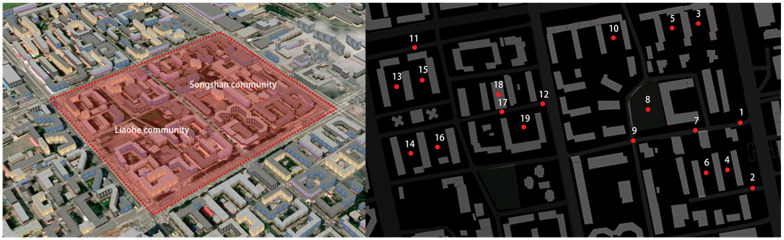
Layout of the study area and measurements of the points set.

**Figure 2 ijerph-19-07212-f002:**
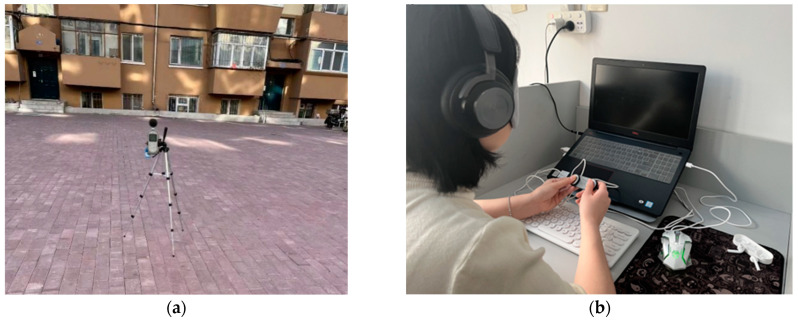
Test instruments: (**a**) acoustic environment test and (**b**) physiological measurement.

**Figure 3 ijerph-19-07212-f003:**
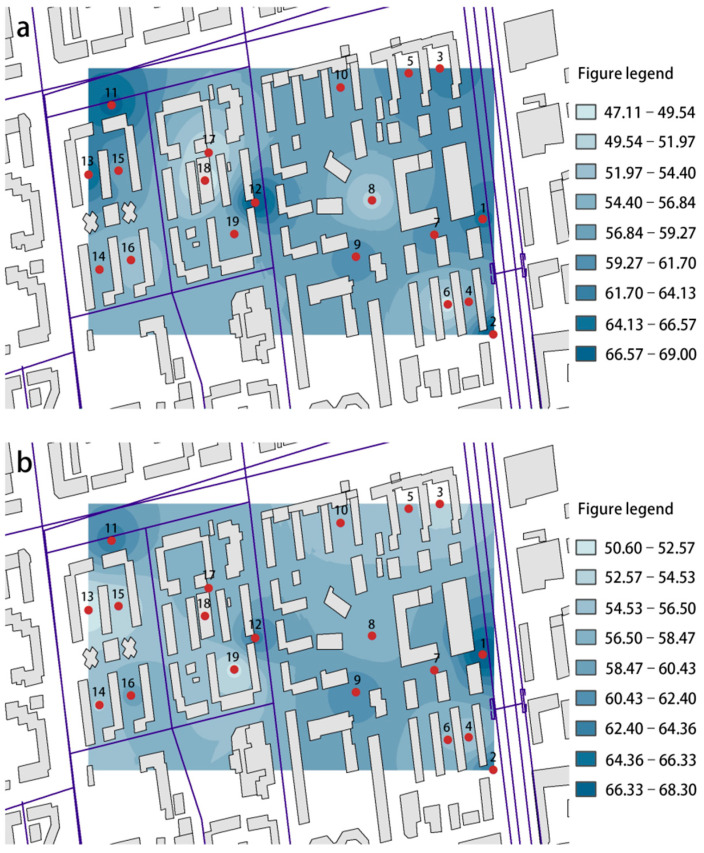
The SPL distribution of the study area during different periods (Unit: dB(A)): (**a**) 8:00–10:00, (**b**) 14:00–16:00, and (**c**) 19:00–21:00.

**Figure 4 ijerph-19-07212-f004:**
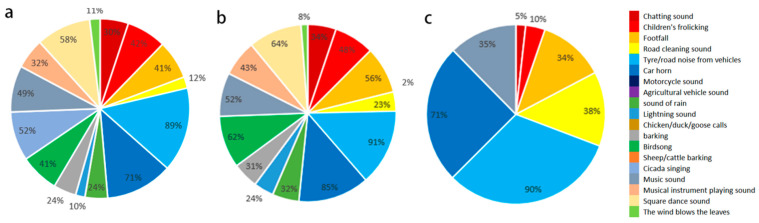
Outdoor sound frequency diagram of an old residential area: (**a**) summer; (**b**) spring and autumn; (**c**) winter.

**Figure 5 ijerph-19-07212-f005:**
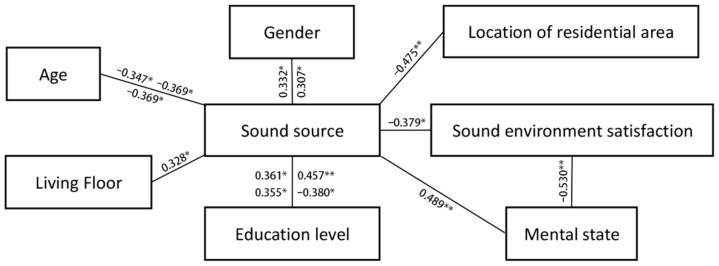
The relationship between different variables and the comfort evaluation of the acoustic environment (** indicates that the two-tailed test is significant at the 0.01 level, and * indicates that it is significant at the 0.05 level).

**Figure 6 ijerph-19-07212-f006:**
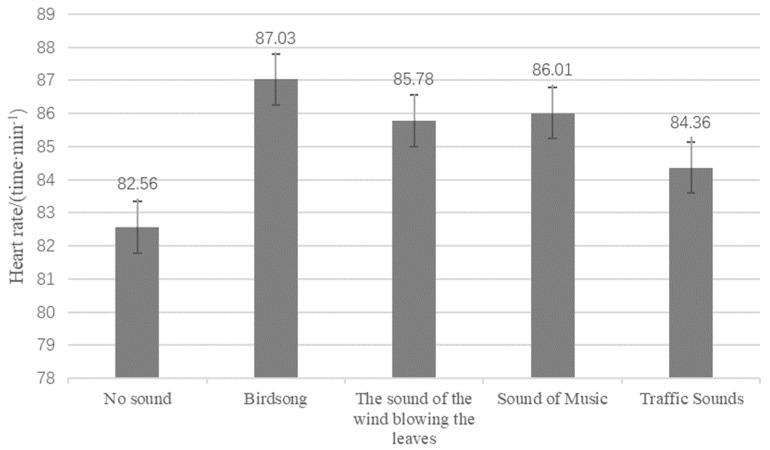
Heart rate changes after different types of sound stimulation.

**Figure 7 ijerph-19-07212-f007:**
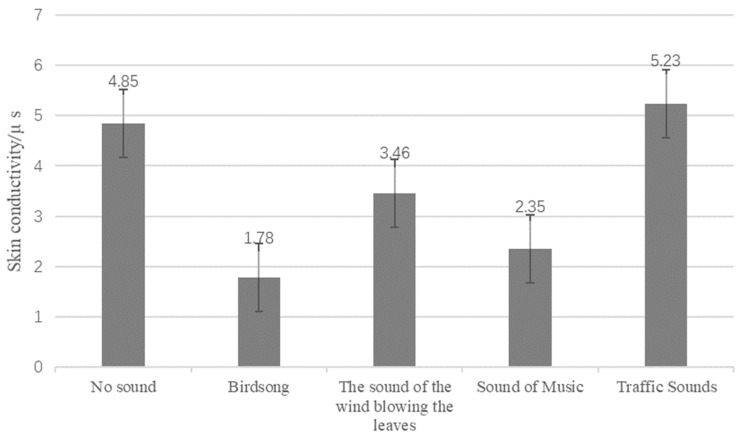
Changes in SCL after different types of sound stimulation.

**Figure 8 ijerph-19-07212-f008:**
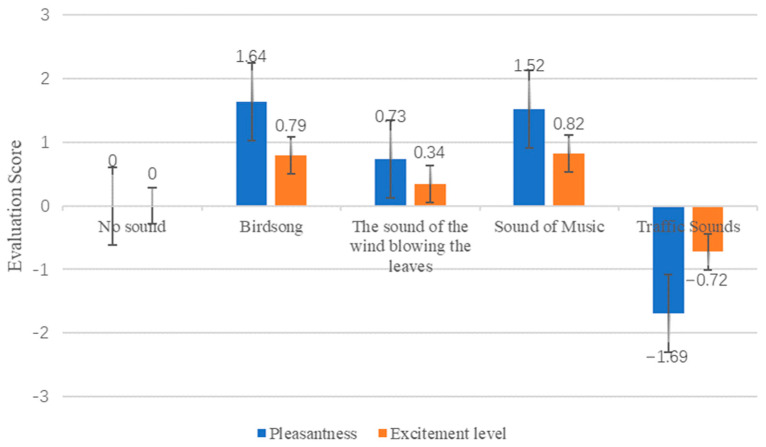
Changes in pleasantness and excitement.

**Figure 9 ijerph-19-07212-f009:**
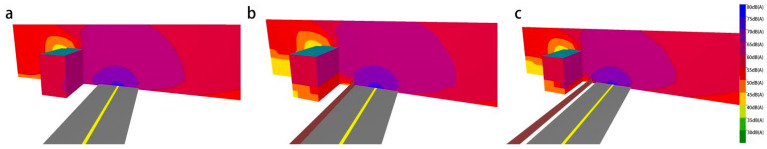
Cross-section simulation of street-facing building noise: (**a**) without acoustic barrier; (**b**) with an acoustic barrier; (**c**) with an acoustic barrier at 2 m.

**Table 1 ijerph-19-07212-t001:** Contents of the questionnaire.

Category	Questions	Scale
Sound source	What kind of sound can you hear now?	
Degree of sound preference	Voice	Enjoy: 5
	Traffic sounds	Like: 4
	Machine sounds	General: 3
	Construction sounds	Dislike: 2
	Musical sounds	Hate: 1
Satisfaction level of sound environment	Sound environment	Very comfortable: 5
		Comfortable: 4
		General: 3
		Uncomfortable: 2
		Very uncomfortable: 1
Background	Gender, Age, Education level, Occupation, Income (per month)	
	Residential location, Residential floor, Length of residence	
PHQ-9 Depression	1. Work with little enthusiasm or interest	0: Never 1: Several days 2: Half 3: Always
	2. Feeling down, depressed, or hopeless	
	3. Difficulty falling asleep, restlessness, or excessive sleep	
	4. Feeling tired or without energy	
	5. Feeling you’re a failure, or you’ve let yourself or your family down	
	6. Have trouble focusing on things	
	7. Move or speak slowly enough for others to notice? Or just the opposite, fidgety or fidgeting and moving more than usual	
	8. Loss of appetite or eating too much	
	9. Suicide or want to harm yourself	

**Table 2 ijerph-19-07212-t002:** Contents of the questionnaire.

Item	Adjective	Value					Adjective	Description
Intelligibility	Blurry	−2	−1	0	1	2	Distinct	Whether the various sound elements can be determined
Perception	Quiet	−2	−1	0	1	2	Bustling	Is the sound environment quiet or bustling
Space	Indoor	−2	−1	0	1	2	Outdoor	Whether the perceived sound occurred indoors or outdoors
Time	Nighttime	−2	−1	0	1	2	Daytime	Whether the perceived sound occurred at nighttime or daytime
Complexity	Simple	−2	−1	0	1	2	Complex	Whether the composition of the sound is complex
Fluency	Harsh	−2	−1	0	1	2	Smooth	Does the sound flow smoothly
Character	Artificial	−2	−1	0	1	2	Natural	Whether the sound sounds natural
Attraction	Unappealing	−2	−1	0	1	2	Attractive	Whether the sound is attractive
Atmosphere	Boring	−2	−1	0	1	2	Dynamic	Whether the sound reflects the environment
Emotion	Placid	−2	−1	0	1	2	Excited	After hearing the sound, there is inner peace or excitement
Mood	Sorrowful	−2	−1	0	1	2	Pleasant	After hearing the sound, the mood is sad or happy

**Table 3 ijerph-19-07212-t003:** The maximum value of SPL during the different periods in the study area (Unit: dB(A)).

Time	8:00–10:00	14:00–16:00	19:00–21:00
Point 1	80.6	74.3	86.0
Point 2	86.7	84.1	78.2
Point 3	76.5	60.2	67.8
Point 4	72.2	65.3	63.1
Point 5	69.4	66.5	68.8
Point 6	58.4	64.8	65.8
Point 7	67.2	67.9	73.6
Point 8	66.3	64.3	71.7
Point 9	70.3	68.3	78.6
Point 10	65.7	60.3	70.6
Point 11	74.8	69.4	91.5
Point 12	84.4	75.9	78.2
Point 13	74.8	60.5	70.8
Point 14	69.1	64.3	67.2
Point 15	67.9	65.6	68.0
Point 16	60.5	67.1	65.5
Point 17	59.7	69.3	70.3
Point 18	58.3	68.6	70.4
Point 19	63.6	66.2	67.6

**Table 4 ijerph-19-07212-t004:** Evaluation of outdoor sound preferences.

Sound Types	Sound Sources	Value
Activity sound	Voices chatting	2.48
	The sound of children playing	3.15
	Clop-clop sound	2.50
	Road sweeping	2.52
	Firecrackers	2.81
Traffic noise	Tire/road noise of traffic	2.05
	Car horn	2.14
	Motorcycle sound	2.09
Mechanical sound	Construction sounds	1.69
	Factory machinery	1.61
	Device running sound	1.82
Background music	Rustling leaves	3.94
	Wind and rain	3.01
	Thunder	2.15
Animal sound	Poultry twitter	2.39
	Barking	2.56
	Birdsong	4.10
	Cicada chirp	2.77
Background music	Musical sound	3.55
	Instrumental sound	3.58
	Square dance sound	3.23

**Table 5 ijerph-19-07212-t005:** Correlation analysis of psychological and physiological index changes.

Item	HR	EDA
Pleasantness	0.268 **	−0.310 **
Excitement	0.168 **	−0.179 **

Note: ** indicates that the two-tailed test is significant at the 0.01 level, and * indicates significance at the 0.05 level.

**Table 6 ijerph-19-07212-t006:** SD scores for different types of sounds.

Number	Adjective		Birdsong	Musical Sound	Rustling Leaves
1	Blurry	Distinct	1.43 **	1.11 **	1.33
2	Quiet	Bustling	0.42 **	0.49 **	0.03 *
3	Indoor	Outdoor	1.90 **	1.30	1.74 **
4	Nighttime	Daytime	1.28 **	1.33	1.11
5	Simple	Complex	1.05	1.39 **	1.53
6	Harsh	Smooth	1.40 **	1.38 **	1.1
7	Artificial	Natural	1.65 **	1.74 *	1.69 **
8	Unappealing	Attractive	1.32 **	0.97 **	1.3
9	Boring	Atmosphere	1.43 **	1.11	1.24 **
10	Placid	Excited	0.44 *	0.48	0.49 *
11	Sorrowful	Pleasant	1.39 **	1.34 **	1.08

Note: ** indicates that the two-tailed test is significant at the 0.01 level, and * indicates significance at the 0.05 level.

**Table 7 ijerph-19-07212-t007:** Numerical change of SPL for each layer after adding an acoustic barrier (Unit: dB(A)).

Floor	Without Acoustic Barrier (a)	With Acoustic Barrier (b)	With Acoustic Barrier at 2 m (c)	SPL Difference between (a) and (b)	SPL Difference between (a) and (c)	SPL Difference between (b) and (c)
1	63	50	50	13	13	0
2	63	55	62	8	1	7
3	63	62	63	1	0	1
4	63	62	63	1	0	1
5	62	61	62	1	0	1
6	62	62	62	0	0	0

**Table 8 ijerph-19-07212-t008:** Changes of SPL in each layer of the sound barrier under different setback conditions (Unit: dB(A)).

Floor	8 m	18 m	28 m	38 m	48 m	58 m
1	63	59	56	54	53	52
2	63	61	58	56	54	53
3	63	61	59	57	55	54
4	63	61	59	58	56	55
5	62	61	59	58	57	55
6	62	60	59	58	57	56
Value	376	363	350	341	332	325

**Table 9 ijerph-19-07212-t009:** Numerical changes of SPL for each layer after adding sound barriers under different setback conditions (Unit: dB(A)).

Floor	8 m	18 m	28 m	38 m	48 m	58 m
1	50	49	47	46	45	44
2	62	52	50	48	46	45
3	63	60	54	49	48	46
4	63	60	58	57	48	47
5	62	61	58	57	56	50
6	62	60	58	57	56	55
Value	362	342	325	314	299	287

## Data Availability

The original contributions presented in the study are included in the article; further inquiries can be directed to the corresponding author/s.

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
