# Peer review of "Research on the Effects of Soundscapes on Human Psychological Health in an Old Community of a Cold Region"

_ijerph, 2022, doi:10.3390/ijerph19127212_

Round 1

Reviewer 1 Report

The manuscript reports the results of a study on the soundscape of an urban area. Different methodologies were used for the evaluation: on-site questionnaires, recordings (both perceptual assessment and physiological measures), and finally numerical simulations of a possible design strategy.

I fully appreciate the first part of the study and in particular the investigation of the relationship between physiological measures and perceptual results, which shows a close link between the two aspects. However, I am not convinced by the last part of the study that (i) should not be presented as a Discussion, and (ii) reduces the design of the sonic environment to the control of the sound level, without considering a “real” soundscape design as one would expect given the first part of the study. Also, the manuscript is missing a proper Discussion section, resuming and critically analyzing the study results.

On these bases, I deem that the manuscript should undergo a major revision before being considered for publication.

Author Response

We really appreciate the valuable suggestion you give. We rewrite the Discussion part in the revised version. For question ii, the second part of the design strategy is related to the previous part of the research. Firstly, the paper obtains the subjects' perception of the existing acoustic environment, and puts forward the corresponding improvement strategies based on the perception. However, this paper takes the old community as the carrier which is different from other research carriers due to the impact of the built-up environment, it can only regulate the background sound and noise. However, through the construction of sound barrier of trees, it can improve the ecological environment and increase the residence time of birds. Factors affecting sound comfort, such as firecrackers and children's noise, can be improved through property management and control. Based on your valuable suggestions, we have added the filtering content of soundscape design elements in the modified version.

Reviewer 2 Report

 Cui et al. investigated the effects of Soundscapes on Human Psychological Health. They did a prior survey as well as the soundscapes measurement. 

The results show that the traffic noise and other unwanted noise had a negative impact on residents. In contrast, the birds' chirping had a positive impact.  The findings are obvious. I have some suggestions and concerns for this study.

> How significantly cars parking in a disorderly fashion, and illegal buildings, affect the dissipation of noise. Kindly provide some reference if any such studies have been reported previously.

> Since this study involves human participants, please write a paragraph if any prior permission was taken from respective regulators.

>Mention the simulation software used for this study. Also, provide a brief description of the simulation process. 

> Was the voice recorder calibrated as per ISO/ASME standards?

> Te manuscript states that sound was recorded at relatively quiet times throughout the day. It's a qualitative thing! Were there any criteria used for selecting the time? Because a change in SPL values can significantly influence the consistency of the recording data. A range should be mentioned.

> The simulated results are discussed in the manuscript properly. 

Author Response

> How significantly cars parking in a disorderly fashion, and illegal buildings, affect the dissipation of noise. Kindly provide some reference if any such studies have been reported previously.

Thank you for your suggestion. We add some discussion in 3.1, 3.2 and Discussion part in the revised version.

> Since this study involves human participants, please write a paragraph if any prior permission was taken from respective regulators.

Yes, we have provided the relevant contents to the journal when the paper is submitted.

>Mention the simulation software used for this study. Also, provide a brief description of the simulation process. 

Thank you for your valuable suggestion. We add the simulation introduction in the revised version.

> Was the voice recorder calibrated as per ISO/ASME standards?

Yes, we calibrated the recorder according to ISO 1996-2:2017, and mentioned in the revised version.

> Te manuscript states that sound was recorded at relatively quiet times throughout the day. It's a qualitative thing! Were there any criteria used for selecting the time? Because a change in SPL values can significantly influence the consistency of the recording data. A range should be mentioned.

Thank you for your valuable suggestion. We add the description to this section in the revised version. (Line: 268-272)

> The simulated results are discussed in the manuscript properly. 

Thank you for your valuable suggestion. We rewrite the Discussion part in the revised version.

Round 2

Reviewer 1 Report

The Authors answered in a satisfactory manner to all my concerns.

Reviewer 2 Report

There is a significant improvement in content in the revised version and the authors have incorporated the suggestions from all the reviewers. It can be considered for the next stage.